# Comparison of Different Label-Free Techniques for the Semi-Absolute Quantification of Protein Abundance

**DOI:** 10.3390/proteomes10010002

**Published:** 2022-01-07

**Authors:** Aarón Millán-Oropeza, Mélisande Blein-Nicolas, Véronique Monnet, Michel Zivy, Céline Henry

**Affiliations:** 1PAPPSO, Micalis Institute, INRAE, AgroParisTech, Université Paris-Saclay, 78350 Jouy-en-Josas, France; veronique.monnet@inrae.fr (V.M.); celine.henry@inrae.fr (C.H.); 2PAPPSO, INRAE, CNRS, AgroParisTech, GQE-Le Moulon, Université Paris-Saclay, 91190 Gif-sur-Yvette, France; melisande.blein-nicolas@universite-paris-saclay.fr (M.B.-N.); michel.zivy@inrae.fr (M.Z.)

**Keywords:** label free, metabolic models, *Saccharomyces*, semi-absolute quantification, quantitative proteomics, TPA, UPS2

## Abstract

In proteomics, it is essential to quantify proteins in absolute terms if we wish to compare results among studies and integrate high-throughput biological data into genome-scale metabolic models. While labeling target peptides with stable isotopes allow protein abundance to be accurately quantified, the utility of this technique is constrained by the low number of quantifiable proteins that it yields. Recently, label-free shotgun proteomics has become the “gold standard” for carrying out global assessments of biological samples containing thousands of proteins. However, this tool must be further improved if we wish to accurately quantify absolute levels of proteins. Here, we used different label-free quantification techniques to estimate absolute protein abundance in the model yeast Saccharomyces cerevisiae. More specifically, we evaluated the performance of seven different quantification methods, based either on spectral counting (SC) or extracted-ion chromatogram (XIC), which were applied to samples from five different proteome backgrounds. We also compared the accuracy and reproducibility of two strategies for transforming relative abundance into absolute abundance: a UPS2-based strategy and the total protein approach (TPA). This study mentions technical challenges related to UPS2 use and proposes ways of addressing them, including utilizing a smaller, more highly optimized amount of UPS2. Overall, three SC-based methods (PAI, SAF, and NSAF) yielded the best results because they struck a good balance between experimental performance and protein quantification.

## 1. Introduction

Mass spectrometry-based proteomics has become an essential tool in the study of biological processes because it can provide an overall assessment of the proteomes of organisms, cells, organs, and tissues. Quantitative proteomics has made it possible to estimate the abundance of proteins coming from a given biological source and to compare the results obtained under multiple sets of conditions, with a view to assessing differences in protein features and protein involvement in particular processes or metabolic functions.

Depending on the final objective, proteins may be quantified in relative or absolute terms (as reviewed elsewhere [1]). Relative protein abundance is determined using a label-free shotgun approach, which can provide an overall view of proteomes across multiple situations and can detect thousands of proteins. The latter is possible because the approach is simpler and more versatile than label-based methods (e.g., iTRAQ, iCAT, TMT). However, the label-free shotgun approach cannot be used to compare the relative abundances of different proteins within the same sample because each peptide has unique ionization properties. Consequently, other quantification methods have been developed that can estimate the relative abundance and different biochemical properties of proteins (e.g., size, observable peptides, most intense peptides) [2]. Two methods based on spectral counting (SC)—the protein abundance index (PAI) [3] and the exponentially modified PAI (emPAI) [4]—can be used to compare the levels of different proteins within samples since the values of these indices and molar protein concentrations are positively correlated [4]. However, compared to other quantification methods, SC-based methods, especially emPAI, consistently underperform because they overestimate levels of outlier proteins, notably those that are the most abundant or those with a single peptide spectrum match [5,6]. Another widely used method for comparing proteins within samples is intensity-based absolute quantification (iBAQ) [7], which is equivalent to PAI but uses peptide intensities instead of SC. To date, the accuracy of these methods remains to be evaluated.

In contrast, when proteins are quantified in absolute terms, protein abundances are expressed in clear units (e.g., grams, moles, molecules/cell) and can be easily compared within samples. The absolute quantification method (AQUA) [8,9] is among the most commonly used. It employs isotope-labeled standard synthetic peptides, which are added at known concentrations to cell lysates during digestion to determine the ratio between labeled and unlabeled peptides of interest. Previously used to carry out absolute quantification in mass spectrometry, this technique yields highly accurate measurements of protein abundance. Unfortunately, it has some key drawbacks. It is time-consuming and costly, and, more importantly, it can only quantify a limited number of proteins at a time (in general, less than 100) [10,11]. The latter issue is quite limiting in the context of systems biology, given that absolute quantification is needed for large proteomics datasets (i.e., covering 1000s of proteins) if they are to be properly integrated into genome-scale metabolic models, where protein abundances may serve as upper limits [12,13]. In addition, by making available more proteomics datasets that use absolute values, it would be easier to carry out comparative studies across laboratories even when data have been acquired with different devices.

Label-free absolute quantification is therefore useful because it reduces the cost and complexity of sample preparation and allows larger numbers of proteins to be quantified. Indeed, all the peptides detected can be quantified, unlike in methods employing isobaric tags (iTRAQ, TMT) [14,15,16] that only quantify labeled peptides. Moreover, in label-free methods, there is no limit on the number of samples per experiment, which is not true for commercial label-based methods (e.g., TMTpro^TM^ 16plex Label Reagent Set, Thermo Scientific, Rockford, IL, USA).

There are two main label-free strategies for transforming the unitless measurements of intensity provided by the mass spectrometer into moles per gram of material. In the current terminology, the use of these strategies results in semi-absolute quantification. First, there is the total protein approach (TPA) [17], which is rooted in the assumption that the total mass spectrometry signal (based on SC or extracted-ion chromatogram (XIC)) for all the proteins in a given sample reflects the total amount of protein present. Hence, the signal for a given protein should be proportional to its true abundance in the sample. This strategy has made it possible to carry out semi-absolute quantification without the need for external standards [18]. However, the protein abundances calculated using TPA might be wide of the mark since, in general, more than 60% of peptide fragments are not assigned at the protein identification stage. While numerous studies have applied various forms of TPA [19,20,21,22,23], it remains to be determined whether this strategy accurately estimates protein abundance. Second, there is a commonly employed strategy that relies on an external standard, the Universal Proteomics Standard 2 (UPS2). In this strategy, UPS2 proteins are added in known amounts to establish a standard of reference with which unitless intensities can be converted into concrete abundances. UPS2 contains a mixture of 48 human proteins at six different molar concentrations, where there are eight proteins of different molecular masses present at each concentration level. Multiple studies have found strong positive correlations between the expected and observed relative abundances of UPS2 proteins [7,24,25,26,27,28,29,30,31]. However, in all cases, there was a need for massive amounts of UPS2 to carry out quantification (e.g., up to 3–10 µg per mass spectrometry run), which is a constraint if the goal is to analyze large cohorts. Indeed, UPS2 is costly and is not available year-round. Furthermore, there is no consensus on either the ratio (protein standard/proteome background) or the amount of UPS2 required to develop a suitable standard of reference. However, there is a clear need to optimize this strategy by reducing the amount of UPS2 needed while maximizing the number of proteins that can be detected.

This situation, therefore, calls for the development of a practical, robust label-free technique for carrying out semi-absolute protein quantification that will yield large proteomics datasets of sufficient accuracy to be integrated into metabolic models. To this end, we evaluated fourteen different label-free semi-absolute quantification techniques (seven quantification methods × two transformation strategies). Our study system was chemostat cultures of the budding yeast *Saccharomyces* cerevisiae (CEN.PK113-7D) cultivated under five different sets of conditions.

## 2. Materials and Methods

### 2.1. Yeast Cultures

Cultures of *Saccharomyces cerevisiae* CEN.PK113-7D (Dr P. Kötter, Frankfurt, Germany) were grown in 500-mL chemostats at a dilution rate of 0.1 h^−1^. We used a synthetic medium containing 5 g/L of (NH_4_)_2_SO_4_ (Sigma-Aldrich, Saint Louis, MO, USA), 3 g/L of KH_2_PO_4_ (Sigma-Aldrich, Saint Louis, MO, USA), 0.5 g/L of MgSO_4_·7H_2_O (Sigma-Aldrich, Saint Louis, MO, USA), 7.5 g/L of glucose (Sigma-Aldrich, Saint Louis, MO, USA), trace elements (VWR, Fontenay-sous-Bois, France), vitamins (VWR, Fontenay-sous-Bois, France), and 1 g/L of pluronic PE6100 (Parchem, New Rochelle, NY, USA) to reduce foaming. Cultures experienced one of five different sets of conditions: standard (30 °C, pH = 5.5), low pH (30 °C, pH = 3.5), high temperature (36 °C, pH = 5.5), osmotic stress (30 °C, pH = 5.5, 1 M KCl), and anaerobic. Each set of conditions was independently replicated three times, resulting in a total of 15 samples (5 conditions × 3 replicates). Sampling was carried out as described elsewhere [32]. Here, we used five different stress conditions of the model yeast *S. cerevisiae* to assess the semi-absolute protein quantification in a realistic dataset, with multiple phenotypes. This motivation aimed to identify the impact of the different quantification techniques on the peptides and proteins abundance trends observed between the different stress responses.

### 2.2. Total Protein Extraction and In-Gel Digestion

The protein preparation protocol is described in the Appendix A. A 5-µg aliquot of total proteins was taken from each of the 15 yeast samples; these aliquots were then pooled to generate a representative sample (hereafter, “bulk” sample; Figure 1A). Then, we took 15 µg of total proteins from each of the 15 samples and 6 different bulk samples (the motivation of this choice is explained in the next section) and separated out their peptides on one-dimensional SDS-PAGE short-migration gels (1 × 1 cm lanes, NP321BOX, Invitrogen, Carlsbad, CA, USA). Details on the digestion process are available in the Appendix A. Extracted tryptic peptides were vacuum dried and resuspended in 75 μL of loading buffer containing 0.08% (*v*/*v*) of trifluoroacetic acid (TFA) (Thermo Fisher Scientific, Rockford, IL, USA) and 2% (*v*/*v*) of acetonitrile (ACN) (Biosolve, Valkenswaard, The Netherlands) in water. The result was digested peptide mixtures at a concentration of 200 ng/µL. Subsequently, 4 µL of each mixture was analyzed using a high-resolution mass spectrometer.

### 2.3. Preparation of the UPS2 Samples

UPS2 (Sigma-Aldrich, Saint Louis, MO, USA) contains 48 human proteins with different molecular masses (6–83 KDa) at 6 different molar concentrations, ranging from 500 amoles to 50 pmoles. The contents of one vial of UPS2 (10.6 µg) underwent reduction, alkylation, and digestion as described in the Appendix A. The extracted tryptic peptides were vacuum dried and resuspended in 25 μL of loading buffer. We took 1.5-µL samples from the mixture of UPS2 digested peptides (424 ng/µL) and spiked them into 7.5 µL of each of the six bulk samples (200 ng yeast peptides/µL) at a ratio of 1:2.35 (UPS2:yeast). Then, 4 µL of this mixture (949 ng) was analyzed using a high-resolution mass spectrometer (Thermo Fisher Scientific, San Diego, CA, USA) (Figure 1A). These bulk samples were thus used as references when there was spiking with the digested UPS2 (Sigma). In this fashion, six replicates of UPS2-spiked bulk samples were randomly analyzed over the course of the experiment, and there was no need to spike each experimental sample with UPS2. This approach considerably reduced the quantity of UPS2 required to obtain reliable correlation results.

### 2.4. Sample Preparation for Method Validation

An experiment was conducted to validate the quantification techniques. Additional bulk samples containing 15 µg of yeast proteins (four independent replicates) were supplemented with purified enzymes at different concentrations: insulin (Sigma, ref. I5500, Saint Louis, MO, USA) at 51.9 fmol; alpha-lactalbumin (Sigma, ref. L5385, Saint Louis, MO, USA) at 108.6 fmol; myoglobin (Sigma, ref. M0630, Saint Louis, MO, USA) at 181.8 fmol; and ribonuclease A (Sigma, ref. R5500, Saint Louis, MO, USA) at 342.6 fmol. The samples underwent in-gel digestion as described above. The extracted tryptic peptides were vacuum dried and resuspended in 75 μL of loading buffer. Then, 1.5 µL of digested UPS2 (424 ng/µL) was spiked into 7.5 µL of the digested peptides (from the yeast proteins and purified enzymes). The expected and observed protein abundances for the samples were compared for the different quantification techniques.

### 2.5. Mass Spectrometry Analysis

Mass spectrometry was performed using a Dionex U3000 RSLC system coupled to an Orbitrap Fusion™ Lumos™ Tribrid™ mass spectrometer (Thermo Fisher Scientific, San Diego, CA, USA). Four μL of digested peptides were injected for each sample. A description of the liquid chromatography procedure and the MS/MS method can be found in the Appendix A.

### 2.6. Protein Identification

To identify the proteins, we used a custom-made database (see the Appendix A), which contained genomic information for *S. cerevisiae* CEN.PK113-7D (UNIPROT, accessed on 14 September 2017) and *S. cerevisiae* s288c (accessed on UNIPROT, 3 November 2017); the UPS protein database (Sigma); and a database of common contaminants. Database searches were performed using the X!Tandem algorithm (Alanine 1 February 2017) implemented in the open-source search engine X!TandemPipeline (v. 3.4.3; https://forgemia.inra.fr/pappso/xtpcpp (accessed on 5 January 2022)) [33]. Enzymatic cleavage was defined as trypsin digestion with one possible miscleavage. The settings for the carboxyamidomethylation of cysteine residues and the oxidation of methionine residues were static modifications and potential modifications, respectively. Precursor and fragment mass tolerance was 10 ppm. The data filtering standards were as follows: peptide E-value < 0.01, protein log(E-value) < –3, and a minimum of one identified peptide per protein. Using such filtering criteria, the peptide and protein false discovery rates (FDR) were 0.04% and 0.68%, respectively. The mass spectrometry data were deposited online in the public database PROTICdb [34,35,36] (repository: http://moulon.inra.fr/protic/chassy_saccharomyces_absolute (accessed on 5 January 2022)) and in the Proteomics Identification Database [37] (PRIDE; dataset identifiers PXD014765 and PXD012836). PRIDE is a member of the ProteomeXchange Consortium.

### 2.7. Protein Quantification

Protein abundance was quantified using SC- and XIC-based methods (Figure 1B). The R code that we used is available at https://forgemia.inra.fr/aaron.millan-oropeza/protquanter-saq/ (accessed on 5 January 2022). Four SC-based methods were employed. The first was the protein abundance index (PAI), in which the observed number of peptides is divided by the expected number of peptides [3]. The second was the exponentially modified PAI (emPAI), which is equal to 10^PAI minus one [4]. The third was the spectral abundance factor (SAF), in which the SC for a given protein is divided by the protein’s length (*L*). The fourth was the normalized spectral abundance factor (NSAF), which is defined for a given protein *i* as follows:(1)$$NSAF_{i}=\frac{{\left(SC/L\right)}_{i}}{\sum_{j=1}^{N}\left(SC/L\right)}$$
where the number of SCs identified for protein *i* is divided by the protein’s length (*L*) in amino acids, and the result is then divided by the sum of *SC*/*L* for all the proteins (*N*) in the experiment [38].

For the XIC-based methods, XIC extraction was carried out using MassChroQ [39] (v. 2.2.16, PAPPSO, Gif-sur-Yvette, France); the peak detection threshold was between 30,000 and 50,000, and the range was 10 ppm. The resulting peptide intensities (i.e., the areas under the curve) were log10-transformed before carrying out further data analyses. We eliminated peptides when their standard deviation of retention time was higher than 20 s. A local normalization method, described elsewhere [40], was applied. Peptides that belonged to multiple proteins were removed, except in the method in which absolute abundance was intensity-based (iBAQ, described below). Proteins with at least two peptides were kept for further quantification. Three XIC-based methods were used. The first was SUMnorm, in which the sum of the XIC results are normalized by amino acid number [41]. Second was TOP3, in which the mean intensity for the three most intense peptides is associated with a protein [42]. Third was iBAQ, in which the sum of all the MS1 peptide intensities is associated with a protein and divided by the expected number of observable peptides [7].

Two different strategies for transforming relative abundance into absolute abundance were applied to the results of the seven quantification methods: (i) an adapted TPA [17] that takes into account total observed protein abundance (Equation (2)) and (ii) a UPS2-based strategy in which a standard of reference is established using the linear regression between the expected and observed abundances of UPS2 proteins (fmol). Here, the observed abundances were estimated via the different quantification methods (Figure 1B). A total of 70 datasets were generated (7 quantification methods × 2 transformation strategies × 5 sets of culture conditions).
(2)$$Proteinconcentration_{ikm}=\frac{Relativeabundance_{ikm}Totalmass_{k}}{Totalrelativeabundance_{km}MW_{i}}$$
where *relative abundance* is the abundance of protein *i* as estimated by method *m* using sample *k*; *MW* is the molecular weight (g/mol) of protein *i*; *total mass* is the total amount of digested proteins (g) in sample *k*; and *total relative abundance* represents the sum of all the abundances estimated by method *m* using sample *k*. *Protein concentration* is expressed in mol.

### 2.8. Data Analysis

We used R [43] (v. 3.5.1, R Core Team, Vienna, Austria) to quantify protein abundance and carry out the data analyses. The performance of the different techniques (the seven quantification methods × the two transformation strategies) was assessed with various metrics. These metrics were (i) accuracy intra-samples, as estimated by bootstrapping the results for the UPS2 proteins (see the Appendix A for details); (ii) the coefficient of determination (*r^2^*) obtained from the linear regression between the expected and observed abundances of UPS2 proteins (fmol) for a given quantification method; (iii) repeatability, as estimated among all the bulk samples using all the results for the UPS2 proteins (i.e., the CV among replicates); (iv) the variability associated with the results for the UPS2 proteins, which was evaluated for each order of magnitude of concentration and reported as a median value (i.e., the CV among proteins); (v) the number of proteins whose abundance was quantified; and (vi) the ratio between the estimated total mass of all the quantified proteins and the mass of the digested proteins that were actually injected:(3)$$Massratio_{kt}=\frac{\sum_{j=1}^{N}Proteinconcentration_{}MW_{i}}{Totalmass_{k}}$$
where *protein concentration* (mol) is the estimated abundance of a given protein *i* obtained using a given technique *t*; *MW* is the molecular weight (g/mol) of protein *i*; *total mass* is the total mass of digested proteins (g) in a sample *k*; and *N* is the total number of quantified proteins in a sample *k*.

## 3. Results and Discussion

### 3.1. Implementation of the UPS2-Based Strategy in Yeast

To determine the maximum number of UPS2 proteins that we could identify under our experimental conditions, a 400-ng dose of pure UPS2 peptides (i.e., not spiked in a yeast background) was analyzed in a single LC-MS/MS run. A total of 34 out of 48 UPS2 proteins were identified across 5 molar concentrations (ranging from 0.12 fmol to 1200 fmol; Appendix A), which indicates that not all the UPS2 proteins were detectable in our conditions, especially those at the lowest concentration level (0.5 fmol). These results fit with what was observed by Tsou et al. [44], who compared the performance of data-independent acquisition (DIA) methods and data-dependent acquisition (DDA) methods across multiple samples, including a UPS2 sample. Although the injection amount was not specified, it was indicated that the researchers were able to identify 34 proteins after combining three different search engines in the DIA pipeline; however, none of the UPS2 proteins at the lowest concentration level (0.5 fmol) were detected.

Using 1D SDS-PAGE stacking gels, we separated out the peptides in the samples of the *S. cerevisiae* CEN.PK113-7D cultures grown under the five sets of conditions (i.e., standard, low pH, high temperature, osmotic stress, and anaerobic) and in the bulk samples. After in-gel digestion, the UPS2 peptides were spiked into the bulk samples at a ratio of 1:2.4 (UPS2:yeast). This ratio was chosen based on the results of previous assays, in which the use of lower relative quantities of UPS2 (ratios ranging from 1:5 to 1:647, Appendix A) resulted in only small numbers of UPS2 proteins being detected (<15), precluding the ability to perform regression analysis. Here, our UPS2:yeast ratio (1:2.4) resulted in 282 ng of total UPS2 peptides being analyzed and up to 28 UPS2 proteins across five molar concentrations being detected in a single LC-MS/MS run (Appendix A).

Other studies have spiked samples from different organisms with UPS2 before the digestion step. They detected the following number of UPS2 proteins: 32 across four molar concentrations for a full vial of UPS2 (10.6 µg) [24]; 25 across five molar concentrations for a 3.3-µg sample of UPS2 [30]; and 24 across four molar concentrations for a 4.24-µg sample of UPS2 [7]. In these studies, large amounts of UPS2 were used to detect a reasonable number of proteins spanning four orders of magnitude in concentration. Here, we optimized the amount of UPS2 that we used to spike the samples (636 ng, which represents up to 16 times less than in the studies mentioned above) while still aiming to detect an acceptable number of proteins across several molar concentrations. Since only the bulk samples, which served as our references, were spiked with UPS2, the operational costs were considerably reduced. As a result, it is possible to increase the number of bulk samples within an experiment, thus allowing additional experimental samples/conditions to be added.

Based on the recommended amount of UPS2 to be spiked (Figure 1A), the number of bulk samples that can be used climbs from 16 (as obtained from a single UPS2 vial in previous studies) to only two [7], three [30], or ten [26] samples.

### 3.2. Performance of the Quantification Methods with the UPS2 Proteins

The abundances of the 28 UPS2 proteins detected in the bulk samples were quantified using seven different methods (based on SC: PAI, emPAI, SAF, and NSAF; based on XIC: SUMnorm, Top3, and iBAQ; Figure 1B). Prior to carrying out the final regression models, a Cook’s distance analysis [45] was performed to identify any outliers that could negatively affect the linear models. An outlier was defined as a point with a Cook’s distance of more than three times the value of the mean (*μ*). Based on the results of this analysis, the proteins UBIQ (P62988), COS5 (P01031), and SYUG (O76070) were removed from the dataset. The short length of these proteins coupled with the proximity of their arginine and lysine residues resulted in small peptides that were poorly detected after mass fragmentation. In similar studies performed at our proteomics facility using different yeast strains and bacteria (data not shown), COS5 and SYUG were also systematically removed from datasets, suggesting that these proteins cannot be properly quantified via label-free shotgun techniques.

Linear regression analysis was thus carried out using the data for the remaining 25 UPS2 proteins. The relationships between the expected and observed protein abundances for the different quantification methods yielded various standards of reference (Appendix A). Overall, the regression parameters and statistical measures (i.e., intercept, slope, and coefficient of determination) were reproducible across the bulk samples for each quantification method. The six independent bulk sample replicates were employed to establish a final standard of reference that was used to determine the linear regression parameters for estimating the absolute abundance (in fmol) of the yeast proteins. Different metrics were used to evaluate the performance of each quantification method in tandem with the use of this standard of reference (Table 1).

For most of the methods, CV was low among replicates (<11%), with the exception of emPAI (Figure 2). The CV among proteins varied as a function of protein concentration (Appendix A). As expected, the XIC-based methods had higher *r^2^* values for their CVs among proteins and among replicates than did the SC-based methods, the same trend as seen elsewhere [2]. In another study, the CV among replicates ranged from 1 to 16% when AQUA [46] was used. The methods used in this study yielded similar CVs among replicates (Figure 2). Furthermore, the total number of proteins for which absolute abundance could be estimated (2204 for SC-based and 1556 for XIC-based methods) was higher than that obtained with AQUA (<50).

### 3.3. Performance of the Semi-Absolute Quantification Techniques with the UPS2 Proteins

While CV is a useful metric for quantifying repeatability among replicates and proteins, it cannot be used to assess the ability of quantification methods to accurately determine the abundance of a given protein. For this reason, the accuracy of semi-absolute quantification among samples was determined for the different techniques (seven quantification methods × two transformation strategies; Figure 3) by applying an iterative bootstrap to the results for each UPS2 protein (see the Appendix A). Overall, the XIC-based techniques provided more accurate results than did the SC-based techniques. However, when a *t*-test was used to compare the results obtained using SUMnorm and SAF (i.e., the best XIC- and SC-based methods, respectively), no significant difference was found (*p* = 0.209).

### 3.4. Performance of the Semi-Absolute Quantification Techniques with External Proteins

We used an additional experiment to determine which transformation strategy most accurately converted relative abundance into absolute abundance. To this end, we compared the results obtained using the seven quantification methods in tandem with either TPA or the UPS2-based strategy (Figure 4A). In this experiment, bulk samples of yeast were supplemented with purified enzymes: insulin (INS) at 51.9 fmol; alpha-lactalbumin (LALBA) at 108.6 fmol; myoglobin (MYG) at 181.8 fmol; and ribonuclease A (RNAS1) at 342.6 fmol. The mixtures were separated and digested in gel; then, the digested UPS2 peptides were spiked into the samples before determining the absolute abundance of the enzymes (Appendix A).

A rough estimate of overall accuracy was obtained using the total mass ratio (Equation (3)). Using this metric, the XIC-based methods (SUMnorm, TOP3, iBAQ) displayed better performance than the SC-based methods (Table 2). This finding was consistent with the results of the previous experiment (Figure 3). When the transformation strategies were compared, TPA arrived at total mass ratios that were closer to the expected values (mass ratio = 1) than did the UPS2-based strategy. However, this metric should be interpreted carefully since, by definition, TPA calculations are already normalized using total mass.

Absolute error was calculated by comparing the observed and expected abundances of the enzymes. The techniques with the lowest median absolute errors were NSAF (26%), SAF (35%), and PAI (44%) used in tandem with the UPS2-based transformation strategy (Figure 4B). These absolute error values are comparable to those of the commercial kit READYBEADS™ (ANAQUANT, Villeurbanne, France), which is used to quantify absolute abundance with a BSA standard at different concentrations [47]. Another study indicated that TPA could be preferentially employed because it yielded similar results to those of the UPS2-based strategy when used with iBAQ applied to a single yeast sample [19]. This finding fits with the accuracy among samples observed in our study (Figure 3); it contrasts with what we observed for the absolute error of the external proteins (Figure 4B). Indeed, TPA offers the advantage of not relying on external standards.

Taking the results together, the NSAF-UPS2, PAI-UPS2, and SAF-UPS2 techniques displayed a good balance between performance (linearity, reproducibility, accuracy) and the number of proteins quantified. We performed a PCA on the protein abundances (fmol) estimated using the three semi-absolute quantification techniques with the lowest median absolute error with UPS2 approach (Appendix A). In the PCA, the five sets of culture conditions were clearly differentiated (>33% of variance considering the PC1 and PC2), indicating that these quantification techniques could distinguish among the biological features of datasets obtained from different proteome backgrounds. The same pattern was observed for a PCA of the results for the techniques SUMnorm-TPA, iBAQ-TPA, and TOP3-TPA (Appendix A).

Using any of the top-performing SC-based methods (PAI, SAF, NSAF) with the UPS2-based transformation strategy represents a simple way to carry out semi-absolute quantification within a routine shotgun workflow. Nevertheless, the use of XIC-based methods such as iBAQ or SUMnorm with either transformation strategy could prove interesting to implement within new workflows that rely exclusively on peptide intensity data, such as DIA methods.

Based on the previous findings, it could be helpful to replace relative quantification with semi-absolute quantification via the SUMnorm-TPA or iBAQ-TPA techniques in the following situations: (i) the research aims involve the analysis of human proteins, since UPS2 contains human proteins; (ii) UPS2 is not available; or (iii) a large-scale experiment is being carried out that needs cost-effective and straightforward wet-lab procedures.

## 4. Conclusions

In this study, using five different proteome backgrounds, we demonstrated the feasibility of utilizing two transformation strategies—TPA and a UPS2-based approach—as the foundation for label-free semi-absolute quantification. Based on the performance of the fourteen different techniques that we tested, we recommend employing an SC-based quantification method—PAI, SAF, or NSAF—in conjunction with the UPS2-based transformation strategy when calculating protein abundances (fmol) in complex mixtures. The techniques tested in this study remain imperfect, given that the CVs we obtained were inferior to those associated with methods based on isotope labeling (e.g., AQUA, SILAC). However, they do represent a potentially helpful tool for performing semi-absolute quantification in studies where thousands of proteins are being examined and the goal is to generate datasets that can be integrated into metabolic models. Conversely, the XIC-based methods SUMnorm or iBAQ coupled with the TPA transformation strategy could provide a cheaper and simpler alternative for quantifying semi-absolute abundance in large-scale experiments. The proposed techniques could also be employed in fundamental research looking at the metabolic dynamics of microbes, plants, and other organisms. They could be particularly useful in cases where we know little about the stoichiometry of reference proteins, given that such information is needed for absolute quantification.

## Figures and Tables

**Figure 1 proteomes-10-00002-f001:**
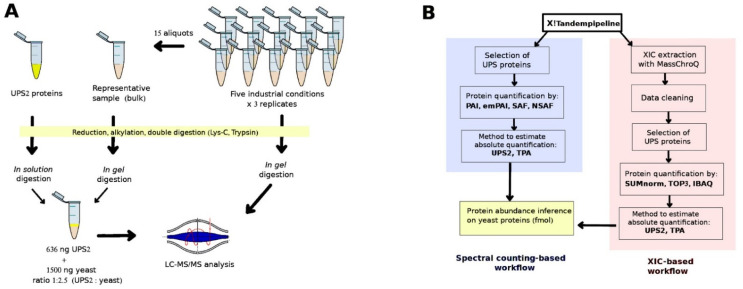
(**A**) Sample preparation and (**B**) the bioinformatic approaches used to determine absolute protein abundances.

**Figure 2 proteomes-10-00002-f002:**
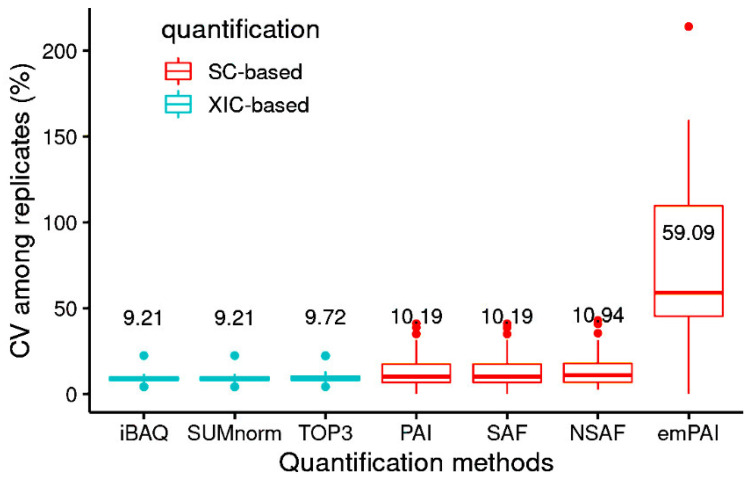
Coefficient of variation (CV) among replicates of the quantification methods (median values; *n* = 6 samples). Median values are indicated above each boxplot.

**Figure 3 proteomes-10-00002-f003:**
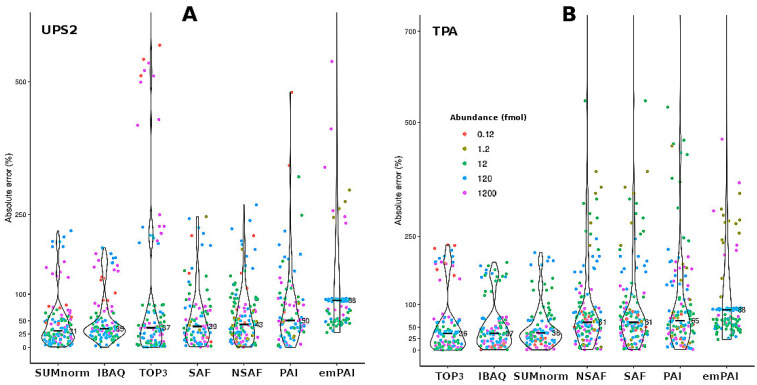
Absolute error among samples for the different quantification methods as obtained by bootstrapping the data for the spiked UPS2 proteins. Results for the two transformation strategies: the UPS2-based approach (**A**) and TPA (**B**). The points are displayed as a function of the different UPS2 protein concentrations (fmol). The inner marks indicate median absolute error (*n* = 144).

**Figure 4 proteomes-10-00002-f004:**
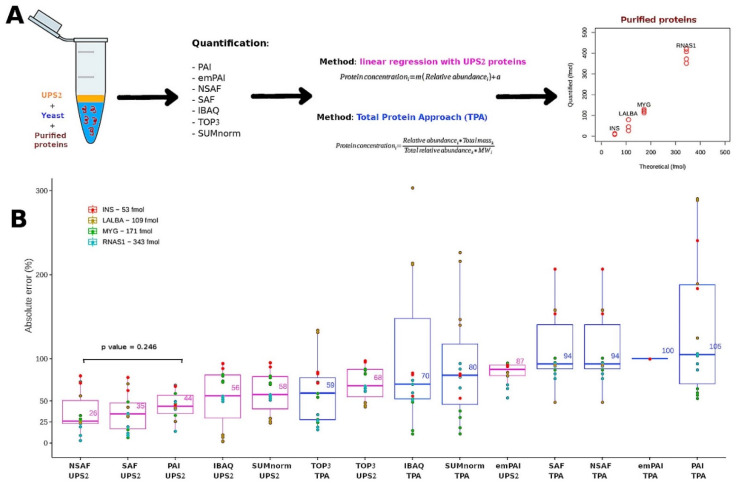
(**A**) Workflows used for validation. For TPA and the UPS2-based strategy, *i* refers to a given protein from a given LC-MS/MS sample *k*. For the standard of reference obtained using the UPS2-based strategy, *m* represents the slope and *a* represents the y-intercept. (**B**) Comparison of the different semi-absolute quantification techniques using the absolute error of the estimated abundance of purified proteins at known concentrations. Results of the *t*-test comparing the NSAF (UPS2) and PAI (UPS2) methods. The UPS2-based strategy is indicated in the pink boxes. TPA is indicated in the blue boxes.

**Table 1 proteomes-10-00002-t001:** Metrics used to evaluate the performance of the quantification methods, based either on spectral counting (SC) or extracted ion chromatogram (XIC). The coefficient of determination (*r^2^*) was calculated from the bulk samples (median values; *n* = 6 samples). The among-protein estimates of coefficient of variation (CV) took into account the variability among UPS2 proteins at all the orders of magnitude of concentration (median values; *n* = 25 proteins).

Quantification Methods		Linearity (*r^2^*)	CV among Proteins (%)	CV among Replicates (%)
SC-based	PAI	0.89	48.8	10.2
	emPAI	0.61	161.4	59.1
	SAF	0.90	48.0	10.2
	NSAF	0.90	48.0	10.9
XIC-based	SUMnorm	0.96	52.9	10.0
	TOP3	0.91	62.6	10.5
	iBAQ	0.96	51.3	10.0

**Table 2 proteomes-10-00002-t002:** Total mass ratio calculated for purified proteins and the spiked UPS2 proteins (median values) using the two transformation strategies (UPS2 and TPA).

Quantification Methods	Purified Proteins		Spiked UPS2 Proteins	
	UPS2	TPA	UPS2	TPA
iBAQ	0.15	0.65	0.69	0.96
SUMnorm	0.16	0.89	0.74	0.95
TOP3	0.21	0.53	1.09	0.96
NSAF	0.21	0.16	1.21	0.92
SAF	0.22	0.16	1.19	0.92
PAI	0.17	0.15	1.17	0.92
emPAI	0.12	3.67	84.83	0.99

## Data Availability

Mass spectrometry data were deposited online in the public database PROTICdb http://pappso.moulon.inra.fr/protic/proticprod/angular/#/projects/199 (accessed on 4 January 2022) and in the Proteomics Identification Database (PRIDE) dataset identifiers PXD014765 and PXD012836.

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
