# Peer review of "Comparison of Different Label-Free Techniques for the Semi-Absolute Quantification of Protein Abundance"

_proteomes, 2022, doi:10.3390/proteomes10010002_

Round 1

Reviewer 1 Report

In their manuscript “Comparison of different label-free techniques for the semi-absolute quantification of protein abundance”, Aarón Millán-Oropeza and co-authors compared 7 developed LC-MS methods for the estimation of absolute protein abundances using a commercial protein standard. The method was originally developed by the Selbach group for the determination of protein levels form mouse samples (DOI: 10.1038/nature10098) and is often used ever since. Comparing different (SC and TIC based) absolute estimation LC-MS approaches has also already been done before and the findings could be largely confirmed in the present study (DOI: 10.1002/pmic.201300135). All this greatly limits the novelty of the presented manuscript. Technically, the authors use very high UPS2 spike-in amounts in their samples, making these proteins the most abundant in their samples. This certainly improves the quantitative performance of the spiked-in proteins, but it considerably decreases overall proteome coverage and increases experiment costs. This could also explain the unexpected good reproducibility and quantitative performance of the SC over the TIC based quantitative methods when, like done in the study, only looking at the UPS2 proteins. The impact of the yeast matrix becomes negligible and as such, with the exception of emPAI, all approaches perform very well. This is clearly not a realistic analytical scenario of a proteome analysis. Even if the performance is reduced, less UPS2 mix should be spiked in and then it would be interesting to see how the different methods perform. Please see more comments below that needs to be addressed before publication:         

Major Comments:

The authors employed different spike-in amount of the rather expensive UPS2 protein standard and found the highest concentration to be the best, which is 1:2.35 (w/w) in yeast. The mean that the 8 most abundant proteins of the UPS2 had concentration of 1.3 pmol on column, whereas the most abundant yeast proteins had amounts of 300-500 fmol, so 3-4x less. This reviewer is just wondering if this high amount of UPS2 is reducing overall coverage of the proteome analysis considering many peptides of the UPS2 standard will dominate the LC-MS analysis. This needs to be tested.

Table 1: This table is a bit misleading. CVs among replicates are the same for SC and XIC (TIC) based methods. I would expect TIC approaches to be much more precise, in particular for lower abundant proteins for which only few peptides are available for quantification. This is also the case when looking at the Box-Plots and the Interquartile ranges. Here, clearly, the XIC based methods outperform SC methods as expected. A plot showing the CVs amount replicates in box-plot format for all 7 methods employed should be included in the main text.

Table 1: The “Proteins per sample” and “Proteins in the whole experiment” should be removed. All previous values are only based on the UPS2 values and then showing the protein numbers of all yeast proteins would lead the reader to think that the previous columns are based on identified yeast proteins, which is not the case.  

Figure 3: The errors are just based on 4 different proteins with some clearly showing very strong errors in a consistent manner. Thus, even if replicates are performed, the 4 proteins hardly seems representative to draw general conclusions. Also, the dynamic range from 52 to 342 fmol does not even cover one order of magnitude and is obviously biased towards higher abundant proteins. This (main) validation experiment should be repeated with at least 10 proteins covering 1 pmol to 1 fmol and also including larger proteins, like BSA. This should allow the generation of a more realistic dataset.   

Supplemental Table 1:

Please use “peptide spectrum matches (PSMs)” instead of “spectra” as a header for the PSMs.

Supplemental Table 3:

Please specify protein abundance in more detail than just “in fmol”. It is not clear to what this relates to.

Author Response

Responses to Reviewer 1

In their manuscript “Comparison of different label-free techniques for the semi-absolute quantification of protein abundance”, Aarón Millán-Oropeza and co-authors compared 7 developed LC-MS methods for the estimation of absolute protein abundances using a commercial protein standard.

The method was originally developed by the Selbach group for the determination of protein levels form mouse samples (DOI: 10.1038/nature10098) and is often used ever since. Comparing different (SC and TIC based) absolute estimation LC-MS approaches has also already been done before and the findings could be largely confirmed in the present study (DOI: 10.1002/pmic.201300135). All this greatly limits the novelty of the presented manuscript. Technically, the authors use very high UPS2 spike-in amounts in their samples, making these proteins the most abundant in their samples. This certainly improves the quantitative performance of the spiked-in proteins, but it considerably decreases overall proteome coverage and increases experiment costs. This could also explain the unexpected good reproducibility and quantitative performance of the SC over the TIC based quantitative methods when, like done in the study, only looking at the UPS2 proteins. The impact of the yeast matrix becomes negligible and as such, with the exception of emPAI, all approaches perform very well. This is clearly not a realistic analytical scenario of a proteome analysis. Even if the performance is reduced, less UPS2 mix should be spiked in and then it would be interesting to see how the different methods perform.

Please see more comments below that needs to be addressed before publication:         

Major Comments:

  • The authors employed different spike-in amount of the rather expensive UPS2 protein standard and found the highest concentration to be the best, which is 1:2.35 (w/w) in yeast. The mean that the 8 most abundant proteins of the UPS2 had concentration of 1.3 pmol on column, whereas the most abundant yeast proteins had amounts of 300-500 fmol, so 3-4x less. This reviewer is just wondering if this high amount of UPS2 is reducing overall coverage of the proteome analysis considering many peptides of the UPS2 standard will dominate the LC-MS analysis. This needs to be tested.

We wish to stress the importance of two parameters to carry out the semi-absolute quantification at the preparative step: i) the ratio ‘proteome :UPS2’ to be spiked (1:2.5 was used in this study) and ii) the amount of injected peptides in the mass spectrometer (636 ng UPS2 were injected per sample in this study).

In this concern, the two classical studies mentioned previously (Selbach’s group, DOI: 10.1038/nature10098 and Schmidt’s group, DOI: 10.1002/pmic.201300135) used considerably higher amounts of UPS2 proteins than those in the present work; with a ratio of 1:3 and 4.24 ug UPS2, and a ratio of 1:5 and 1.06 ug UPS2, respectively.

As stated by the reviewer, we also observed that higher amounts of UPS2 lead to a predominance of the eight most abundant UPS2 proteins, whereas higher amounts of proteome background lead to detection of small numbers of UPS2 proteins, the latter makes difficult to apply the semi-absolute quantification approach.

In this study, we are transparent on the compromise between the suitable ratio (proteome background : UPS2) and the quantity analyzed. In addition, we stated the optimization of reduced UPS2 injected compared to other studies in the last two paragraphs of section 3.1.

Table 1: This table is a bit misleading. CVs among replicates are the same for SC and XIC (TIC) based methods. I would expect TIC approaches to be much more precise, in particular for lower abundant proteins for which only few peptides are available for quantification. This is also the case when looking at the Box-Plots and the Interquartile ranges. Here, clearly, the XIC based methods outperform SC methods as expected. A plot showing the CVs amount replicates in box-plot format for all 7 methods employed should be included in the main text.

The column ‘CV among replicates’ was removed from Table 1 and replaced by a boxplot for all 7 methods employed, as recommended.

Table 1: The “Proteins per sample” and “Proteins in the whole experiment” should be removed. All previous values are only based on the UPS2 values and then showing the protein numbers of all yeast proteins would lead the reader to think that the previous columns are based on identified yeast proteins, which is not the case.  

Columns were removed and a brief sentence was added in the last paragraph of section ‘3.2’

Figure 3: The errors are just based on 4 different proteins with some clearly showing very strong errors in a consistent manner. Thus, even if replicates are performed, the 4 proteins hardly seems representative to draw general conclusions. Also, the dynamic range from 52 to 342 fmol does not even cover one order of magnitude and is obviously biased towards higher abundant proteins. This (main) validation experiment should be repeated with at least 10 proteins covering 1 pmol to 1 fmol and also including larger proteins, like BSA. This should allow the generation of a more realistic dataset.   

We acknowledge this proposal of improvement. However, this experiment can not be repeated due to there following hurdles:

  • the experiments were carried out with a chromatographic column that is not longer in service, neither the type of capillary probe (SilicaTipTM, Emitter, 10 μm, New Objective) due to stock limitations. These factor would lead to a systemic measurement bias because of the stochastic nature of ionization in Mass Spectrometry Analysis.

  • The Mass Spectrometer used for this work is not available during the following weeks to carry out the requested analysis.

Nevertheless, we provide for the reviewer the following graph portraying the distribution of the peptides abundance in the three proteomes of the validation study (yeast, UPS2, and purified proteins/ lab proteins).

Density plot per peptide intensity         Counts of peptides per peptide intensity

As observed, the peptides abundances for the yeast proteome overlaps perfectly in the purified proteins profile as was as for the UPS2 proteome, indicating that the range of the proteins used is suitable for the validation study.

Supplemental Table 1:

Please use “peptide spectrum matches (PSMs)” instead of “spectra” as a header for the PSMs.

Terminology was changed in Table S1.

Supplemental Table 3:

Please specify protein abundance in more detail than just “in fmol”. It is not clear to what this relates to.

A short description was added on each spreadsheet to clarify this data.

Reviewer 2 Report

The manuscript by Oropeza et al. described comprehensive evaluation of different quantification methods used in label free proteomics. Specifically seven different quantification methods (spectral counting and extracted ion chromatogram) were evaluated and described in detail. Overall this study is fairly comprehensive and complete, i have some minor comments and questions.

Authors have used five different protein backgrounds in the study, standard (30 °C, pH=5.5), low pH (30 °C, pH=3.5), high temperature (36 °C, 118 pH=5.5), osmotic stress (30°C, pH=5.5, 1M Kcl), and anaerobic.

Authors need to explain specific reason to use these diverse conditions and what impact these different backgrounds had in terms of protein amount or yield and mass spectrometer data acquisition including number of proteins and peptides identified. 

other suggestions:

on line 40 and 463 ITRAQ should be iTRAQ

Author Response

Responses to Reviewer 2

The manuscript by Oropeza et al. described comprehensive evaluation of different quantification methods used in label free proteomics. Specifically seven different quantification methods (spectral counting and extracted ion chromatogram) were evaluated and described in detail. Overall this study is fairly comprehensive and complete, i have some minor comments and questions.

Authors have used five different protein backgrounds in the study, standard (30 °C, pH=5.5), low pH (30 °C, pH=3.5), high temperature (36 °C, 118 pH=5.5), osmotic stress (30°C, pH=5.5, 1M Kcl), and anaerobic. Authors need to explain specific reason to use these diverse conditions and what impact these different backgrounds had in terms of protein amount or yield and mass spectrometer data acquisition including number of proteins and peptides identified. 

To address this comment, the following text was added to the manuscript at the end of the section 2.1. Yeast cultures:

Here, we used five different stress conditions of the model yeast S. cerevisiae to assess the semi-absolute protein quantification in a realistic dataset, with multiple phenotypes. This motivation aimed to identify the impact of the different quantification techniques on the peptides and proteins abundance trends observed between the different stress responses.

This question was answered accordingly in the second paragraph of page 11:

Taking the results together, the NSAF-UPS2, PAI-UPS2, and SAF-UPS2 techniques displayed a good balance between performance (linearity, reproducibility, accuracy) and the number of proteins quantified. We performed a PCA on the protein abundances (fmol) estimated using the three semi-absolute quantification techniques with the lowest median absolute error with UPS2 approach (Figure S4). In the PCA, the five sets of culture conditions were clearly differentiated (>33% of variance considering the PC1 and PC2), indicating that these quantification techniques could distinguish among the biological features of datasets obtained from different proteome backgrounds. The same pattern was observed for a PCA of the results for the techniques SUMnorm-TPA, iBAQ-TPA, and TOP3-TPA (Figure S4).

other suggestions:

on line 40 and 463 ITRAQ should be iTRAQ

This modification was applied.

Round 2

Reviewer 1 Report

The authors have addressed all my comments satisfactorily and the manuscript is now suited for publication.